# The Suppression of the Epithelial to Mesenchymal Transition in Prostate Cancer through the Targeting of MYO6 Using MiR-145-5p

**DOI:** 10.3390/ijms25084301

**Published:** 2024-04-12

**Authors:** Lee Armstrong, Colin E. Willoughby, Declan J. McKenna

**Affiliations:** Genomic Medicine Research Group, Ulster University, Cromore Road, Coleraine BT52 1SA, UK; armstrong-l16@ulster.ac.uk (L.A.); c.willoughby@ulster.ac.uk (C.E.W.)

**Keywords:** prostate cancer, microRNA, miR-145-5p, epithelial-to-mesenchymal transition, MYO6, biomarker

## Abstract

Aberrant expression of miR-145-5p has been observed in prostate cancer where is has been suggested to play a tumor suppressor role. In other cancers, miR-145-5p acts as an inhibitor of epithelial-to-mesenchymal transition (EMT), a key molecular process for tumor progression. However, the interaction between miR-145-5p and EMT remains to be elucidated in prostate cancer. In this paper the link between miR-145-5p and EMT in prostate cancer was investigated using a combination of in silico and in vitro analyses. miR-145-5p expression was significantly lower in prostate cancer cell lines compared to normal prostate cells. Bioinformatic analysis of The Cancer Genome Atlas prostate adenocarcinoma (TCGA PRAD) data showed significant downregulation of miR-145-5p in prostate cancer, correlating with disease progression. Functional enrichment analysis significantly associated miR-145-5p and its target genes with EMT. MYO6, an EMT-associated gene, was identified and validated as a novel target of miR-145-5p in prostate cancer cells. In vitro manipulation of miR-145-5p levels significantly altered cell proliferation, clonogenicity, migration and expression of EMT-associated markers. Additional TCGA PRAD analysis suggested miR-145-5p tumor expression may be useful predictor of disease recurrence. In summary, this is the first study to report that miR-145-5p may inhibit EMT by targeting MYO6 in prostate cancer cells. The findings suggest miR-145-5p could be a useful diagnostic and prognostic biomarker for prostate cancer.

## 1. Introduction

Prostate cancer is a heterogeneous disease affecting millions of men worldwide, predominantly in regions with a high human development index [1]. Epithelial-to-mesenchymal transition (EMT) is a key biological process required for the progression and metastasis of prostate cancer [2]. EMT is a dynamic process in which epithelial cells are reprogrammed to exhibit a mesenchymal phenotype. The loss of cell polarity and cell–cell adhesion leads to enhanced migratory capacity, invasiveness, and resistance to apoptosis [3]. Multiple factors regulate the EMT process, including several miRNAs [4]. MicroRNAs (miRNAs) are short, non-coding RNA molecules that play an important role in regulating gene expression at the post-transcriptional level.

miRNAs bind to specific sequences in the 3′ untranslated region (3′ UTR) of their target messenger RNAs (mRNAs), resulting in either mRNA degradation or translational repression [5]. By inhibiting the translation of target mRNAs or causing their degradation, miRNAs can effectively silence the expression of specific genes [6]. The small size of miRNAs, typically only 18–25 nucleotides long, allows them to recognize related but not identical target sites, so each miRNA can regulate multiple target mRNAs [7]. Through this sequence-specific interaction, miRNAs provide an important mechanism for fine-tuning the levels of proteins encoded by mRNAs in the cell [8]. The ability of miRNAs to modulate gene expression has implicated them in many biological processes and human diseases, including prostate cancer [9].

Our laboratory has previously demonstrated the role of several miRNAs in prostate cancer, including miR-200c [10], miR-24 [11], miR-210 [12], miR-21 [13], miR-182 [14] and miR-143 [15]. However, there remains a need for further investigation into miRNAs that are specifically associated with the EMT process in prostate cancer [16]. With this objective in mind, miR-145-5p is promising miRNA for further investigation.

The role of miR-145-5p has been studied in various cancers and is generally characterized as a tumor suppressor (reviewed in [17]). Expression of miR-145-5p is downregulated in several cancer types, including bladder [18], breast [19], gastric [20], cholangiocarcinoma [21], clear cell renal cell carcinoma (ccRCC) [22], colorectal [23], oesophageal squamous cell carcinoma (ESCC) [24], oesophageal carcinoma [25], gallbladder carcinoma (GBC) [26], gastric cancer [27], glioma [28], hepatocellular carcinoma (HCC) [29], laryngeal squamous cell carcinoma (LSCC) [30], chronic lymphocytic leukemia (CLL) [31], lung adenocarcinoma (LUAC) [32], lung squamous cell carcinoma (Lung SCC) [33], melanoma [34], non-small cell lung cancer (NSCLC) [35], osteosarcoma [36], retinoblastoma [37], and papillary thyroid carcinoma (PTC) [38].

Given this body of evidence, it is not surprising that emerging research has suggested miR-145-5p also acts as a tumor suppressor in prostate cancer [39]. Various studies have validated multiple gene targets of miR-145-5p in prostate cancer cells, demonstrating its capacity to modulate cell behavior through various signaling pathways [40,41,42]. Previous studies have also demonstrated that overexpression of miR-145-5p can increase the chemosensitivity of prostate cancer cells to first-line chemotherapeutic agents [43,44]. However, the link between miR-145-5p and EMT remains poorly characterized in prostate cancer. Some studies have identified that overexpression of miR-145-5p decreases migration of prostate cells, but have not directly linked miR-145-5p and EMT [45,46,47,48]. The studies that have examined a role for miR-145-5p in EMT have produced contradicting evidence, so there is a clear need for further work in this area [49,50].

Hence, although miR-145-5p has been implicated in EMT and prostate cancer, its precise mechanistic contributions and relevant molecular targets in this context remain poorly defined. Therefore, this study aims to elucidate the functional role of miR-145-5p associated with EMT in prostate cancer by examining the expression profile, downstream effects on putative target genes, and impacts on cell behavior. Additionally, complementary in silico analyses are used to evaluate the clinical significance of miR-145-5p in prostate cancer progression and determine its potential as a prognostic or diagnostic biomarker.

## 2. Results

### 2.1. Reduced miR-145-5p Expression Correlates with Prostate Cancer Progression

qRT-PCR analysis revealed significantly reduced expression of miR-145-5p in the prostate cancer cell lines DU145 and PC3 compared to normal prostate RWPE1 cells (Figure 1a). This reduction in miR-145-5p levels was validated in TCGA PRAD clinical samples, with miR-145-5p expression significantly lower in prostate tumor tissue than normal prostate tissue (Figure 1b). Further in silico analysis of the TCGA PRAD dataset demonstrated an association between decreased miR-145-5p expression and established clinicopathological markers of prostate cancer progression, including increased Gleason score, higher pathological T stage, presence of distant metastasis, and lymph node involvement (Figure 1c–f).

Functional enrichment analysis confirmed that miR-145-5p targets several gene sets that are significantly associated with prostate cancer (Table 1, Appendix A). Additional functional enrichment analysis to help identify the key biological mechanisms involved revealed several significant gene networks associated with EMT and extracellular matrix remodeling. (Table 2, Appendix A).

### 2.2. MYO6 Is a Novel Target of miR-145-5p in Prostate Cancer

The objective of this study was to identify a novel target of miR-145-5p in prostate cancer. Given the tumor suppressive role of miR-145-5p and its association with EMT in other cancers, we aimed to find a target gene known to promote EMT that would be upregulated when miR-145-5p levels are decreased. By cross-referencing known EMT genes, validated miR-145-5p targets, and genes negatively correlated with miR-145-5p expression in TCGA PRAD database, we identified MYO6 as a likely candidate (Appendix A). MYO6 was selected for further analysis due to its previously established link with prostate cancer [51] which we also observed in the functional enrichment analysis performed here (Table 1). The predicted base-pairing between miR-145-5p and the MYO6 transcript was confirmed (Appendix A). Transient overexpression of miR-145-5p in RWPE1, DU145 and PC3 prostate cancer cell lines (Appendix A) consistently led to decreased MYO6 mRNA and protein levels, which was validated by a significant negative correlation between miR-145-5p and MYO6 expression in TCGA PRAD samples (Figure 2). We also demonstrated that MYO6 expression is upregulated in prostate tumors compared to normal tissue, and in higher Gleason grade tumors (Figure 2c). Taken together, these data provide evidence that MYO6 is a direct target of miR-145-5p in prostate cancer. The loss of miR-145-5p expression may therefore promote cancer progression through disinhibition of MYO6 and subsequent induction of EMT.

### 2.3. miR-145-5p Alters Key EMT Markers in Prostate Cell Lines

To validate the role of miR-145-5p in regulating EMT in prostate cells, the expression of several EMT markers were examined following transient knockdown or overexpression of miR-145-5p in RWPE1, DU145, and PC3 prostate cell lines. qRT-PCR analysis revealed that mRNA levels of mesenchymal markers vimentin (VIM), fibronectin 1 (FN1), and actin alpha 2 (ACTA2) were downregulated with miR-145-5p overexpression (Figure 3). Taken together, these results confirm that miR-145-5p modulates the expression of multiple EMT marker genes in prostate cell lines.

### 2.4. miR-145-5p Regulates Proliferation, Migration, and Clonogenic POTENTIAL in Prostate Cell Lines

After confirming that miR-145-5p regulates expression of multiple EMT marker genes, we hypothesized that miR-145-5p would consequently impact prostate cell behaviors relevant to tumor progression. Since miR-145-5p regulates multiple oncogenic targets that drive cell proliferation, we first examined the effects of miR-145-5p modulation on proliferation rates. miR-145-5p inhibition significantly increased proliferation, while miR-145-5p overexpression decreased proliferation in RWPE1, DU145, and PC3 prostate cell lines, as quantified by cell counting assay (Figure 4a).

Given the established role of MYO6 signaling in promoting cell migration, we next assessed if miR-145-5p impacts prostate cell motility. As hypothesized, miR-145-5p overexpression reduced migration of RWPE1, DU145, and PC3 cells, whereas miR-145-5p knockdown enhanced migration of RWPE1, DU145 and PC3 cells, as determined by wound healing assay (Figure 4b, Appendix A). Lastly, we evaluated the effects of miR-145-5p on clonogenic capacity using clonogenic assays. Overexpression of miR-145-5p significantly reduced colony formation, while miR-145-5p inhibition increased colony formation across the tested prostate cell lines (Figure 4c, Appendix A).

### 2.5. Mapping the Functional Network of the miR-145-5p/MYO6 Axis

The effects on cell behavior caused by the manipulation of miR-145-5p reflect the function of MYO6 and the wider regulatory network (Figure 5). The bidirectional network highlights the capacity of miR-145-5p to regulate multiple EMT-associated targets. By modulating EMT through multiple molecular targets and pathways, miR-145-5p can clearly exert greater overall impact than through MYO6 alone. Moreover, miR-145-5p regulates several critical oncogenes including MYC, BRAF, and NRAS, which are known promoters of prostate cancer cell proliferation and clonogenic potential when aberrantly activated.

### 2.6. Potential of miR-145-5p as a Biomarker of Prostate Cancer

Given the association between miR-145-5p levels and prostate cancer progression observed in clinical datasets, we hypothesized that quantification of miR-145-5p may have utility as a diagnostic or prognostic biomarker for prostate cancer. To investigate this, we analyzed miR-145-5p expression and clinical outcome data from TCGA PRAD cohort. Receiver operating characteristic (ROC) curve analysis demonstrated that miR-145-5p expression can effectively discriminate between tumor and normal prostate tissue samples in this cohort (Figure 6a). A combination panel of five miRNAs (miR-221-3p, miR-222-3p, miR-133b, miR-143-3p, and miR-145-5p) improved AUC to 0.977 (Appendix A). Further, patients exhibiting lower miR-145-5p expression showed significantly shorter biochemical recurrence-free survival after initial prostate cancer treatment, as assessed by log-rank test (Figure 6b). Conversely, higher miR-145-5p levels correlated with improved initial and follow-up therapeutic response (Figure 6c,d). For survival analyses, the TCGA PRAD cohort was stratified into quartiles by miR-145-5p expression level (Low < 17.99, High > 18.92 log_2_CPM). Kaplan–Meier plots revealed no significant difference in overall survival, disease-free and progression free intervals (Figure 6e–g), likely due to the limited number of deaths in this cohort. High expression of the miR-145-5p target MYO6 was associated with reduced disease-free survival (Appendix A) but did not meet significance. In summary, our analyses indicate miR-145-5p is a potential diagnostic marker and prognostic indicator of biochemical recurrence in prostate cancer.

Furthermore, since loss of miR-145-5p expression has been consistently linked with many other cancers, it was no surprise to find high AUC values for miR-145-5p in multiple TCGA patient cohorts, suggesting the diagnostic potential of miR-145-5p for detecting cancer (Appendix A). Moreover, combination of miR-145-5p with other miRNAs increased the discriminatory power for detecting prostate cancer (Appendix A). Similarly, the expression levels of both miR-145-5p and MYO6 in tumor tissue is significantly correlated with survival outcomes in several other TCGA patient cohorts, indicating that they could together be useful prognostic biomarkers for different cancers (Appendix A).

## 3. Discussion

Previous studies have explored the function of miR-145-5p in various cancer types and signaling pathways [52,53,54,55,56], but its contribution to EMT in prostate cancer has not been thoroughly characterized. Therefore, we aimed to further investigate the relationship between miR-145-5p and EMT in prostate cancer in this study. This is the first report demonstrating that downregulation of miR-145-5p can promote prostate cancer development through targeting MYO6 and regulating EMT.

We first established that miR-145-5p expression was significantly reduced in prostate cancer cell lines and clinical samples compared to normal controls (Figure 1a,b). Further analysis of clinical samples showed that lower miR-145-5p levels were associated with more advanced cancer stage, as determined by clinicopathological parameters (Figure 1c–f). These findings are consistent with previous studies which found that lower levels of miR-145-5p expression are associated with poorer patient prognosis and treatment response [57,58].

Functional enrichment analysis further validated the link between miR-145-5p and target genes associated with prostate cancer development (Table 1) and EMT (Table 2). Together, these results provide evidence that loss of miR-145-5p is detrimental in prostate cancer, as it leads to reduced regulation of EMT and other critical cellular networks [42,48,49]. However, very few studies have demonstrated a functional connection between miR-145-5p and EMT in prostate cancer thus far. Therefore, we aimed to identify a previously unreported EMT-related target of miR-145-5p in this disease.

By cross-referencing three databases, we strategically filtered potential gene targets for further study. We identified MYO6 as the most promising candidate to investigate, since it appeared in our functional enrichment analyses and appears in multiple EMT gene databases, such as dbEMT2 and EMTome. We experimentally confirmed MYO6 as a miR-145-5p target in vitro in three prostate cell lines, showing that miR-145-5p overexpression significantly reduced MYO6 protein and mRNA levels (Figure 2). We corroborated this in vitro data by analyzing TCGA PRAD data, which revealed a significant negative correlation between miR-145-5p and MYO6 levels, as expected for a direct target interaction. We propose that the upregulation of MYO6 expression observed in tumor versus normal prostate tissue is due, at least in part, to downregulation of miR-145-5p.

Taken together, these results suggest that MYO6 is a direct target of miR-145-5p in prostate cells, as previously shown in gastric cancer [57] and 293T kidney cells [58]. This is significant because others have demonstrated how loss of miR-145-5p leading to increased MYO6 expression can promote EMT and uncontrolled cell growth [52,59]. The MYO6 gene encodes a protein called myosin VI, which is a motor protein involved in intracellular transport and actin cytoskeleton organization [60] Myosin VI regulates the cellular trafficking of cell surface receptors, as well as the adhesion molecules involved in cell–cell and cell-matrix interactions, both of which are critical for maintaining the integrity of epithelial cell layers [61]. Disruption of these interactions is a hallmark of EMT, which may be influenced by over-expression of myosin VI. Myosin VI has also been implicated in the regulation of EMT-associated signaling pathways, such as the TGF-β pathway, thereby presenting another mechanism for myosin VI expression to influence the development of EMT [62].

We confirmed this link by showing that modulating miR-145-5p levels significantly altered expression of EMT marker genes in prostate cells (Figure 3). Furthermore, we demonstrated that inhibiting miR-145-5p markedly increased proliferation, migration, and clonogenicity of prostate cells, while overexpression had the opposite effects (Figure 4). These results further illustrate why miR-145-5p loss is deleterious in prostate cancer, as it enhances the capacity for uncontrolled cell growth. Regarding potential therapeutic applications, our data also show that restoring miR-145-5p levels in prostate cancer cells, thereby reducing MYO6 levels, could be a viable strategy to suppress EMT and inhibit cell growth. Targeting MYO6 has been proposed by others as a potential therapeutic approach in gastric [63] and breast [64] cancers to suppress tumor proliferation and metastasis. A prior study revealed that concurrent overexpression of miR-145-5p and miR-143-3p, a member of the miR-145-5p family, synergistically suppressed MYO6 expression in gastric cancer, exhibiting greater inhibitory effects than miR-145-5p alone [57]. Others have shown that modulation of miR-145-5p expression in prostate cancer cells may be a strategy to confer chemosensitivity [43,44]. However, the effective delivery of small RNA-delivering techniques to tumors remains a considerable barrier to clinical use of this type of targeted therapeutic strategy.

However, it would be overly simplistic to attribute the effects on cell behavior solely to the miR-145-5p/MYO6 interaction. This axis resides at the center of an intricate network of interactions (Figure 5), so the combined effects of these connections ultimately determine miR-145-5p impact on cell phenotype. Prior prostate cancer studies have also shown miR-145-5p can inhibit cell growth via other targets, including WIP1 [45], PLD5 [65], and SOX2 [39]. Other validated oncogenic targets of miR-145-5p include MYC [66,67], BRAF [68] and NRAS [69,70]. While our data suggests miR-145-5p regulation of MYO6 exerts a similar effect in prostate cancer cells, its effect on other targets is also important to acknowledge.

For example, other studies have identified different miR-145-5p targets mediating various cellular effects in different cancer types. A range of in vitro experiments have consistently demonstrated that miR-145-5p can reduce proliferation, invasion, and migration in thyroid cancer [71], non-small-cell lung cancer [72], lung adenocarcinoma [73], liver cancer [74], pancreatic cancer [75], colorectal cancer [76], gastric cancer [77], bladder cancer [78], renal cell carcinoma [79], urothelial cancer [80], breast cancer [81], ovarian cancer [82], chondrosarcoma [83], pleural mesothelioma [84], glioma [85] and invasive glioblastoma [86]. All this evidence points towards the crucial role of miR-145-5p in cancer development and progression, throughout its network of targets.

Given the consistent downregulation of miR-145-5p in tumor tissues, we were interested in its potential utility as a clinically useful biomarker. In general, miRNAs are attractive biomarker candidates because they are much more stably preserved in clinical samples compared to mRNAs and can be readily detected using sensitive and specific PCR-based assays [87]. However, identifying the optimal miRNAs to use for a given disease remains challenging. Our data suggest miR-145-5p expression profiling may be a valuable diagnostic biomarker, as it showed significant ability to distinguish between normal and tumor prostate tissues (Figure 6a). Additionally, a biomarker predicting biochemical recurrence (BCR) following prostate cancer treatment would be clinically useful. BCR is typically defined as a rise in prostate-specific antigen after surgery or radiation, indicating potential tumor regrowth. Here, we found patients experiencing BCR had significantly lower miR-145-5p levels, suggesting it may help predict treatment response (Figure 6b). This corroborates previous work proposing miR-145-5p as a potential prostate cancer biomarker. One study evaluated plasma miR-145-5p as a noninvasive biomarker for prostate cancer screening and diagnosis [88]. The expression of miR-145-5p was measured in 170 patients undergoing prostate biopsy. miR-145-5p levels significantly differed between benign, precancerous, and cancerous prostate pathologies. Multivariate analysis showed miR-145-5p could distinguish between patient groups when combined with clinical assessments. Given our analyses, we agree with the authors’ conclusion that miR-145-5p is a promising plasma biomarker for prostate cancer detection and diagnosis when used with other clinical information. However, further validation on different sample types (e.g., tissue, plasma, serum, urine), is needed in larger, independent patient cohorts.

Others have evaluated miR-145-5p as a valuable adjunct to the Gleason grading system [56]. Due to the heterogeneous and often limited nature of prostate biopsy samples, the Gleason scoring system has inherent variability and limitations in predicting prognosis, even among patients receiving the same Gleason score [89,90]. This study found prostate cancer biopsies that were initially scored Gleason 6 and later upgraded to Gleason 7 after surgery had lower expression of miR-145-5p. The differences in miR-145-5p expression suggests an underlying biological distinction between indolent versus aggressive cancer that appear identical on standard histopathology. Again, this demonstrates the potential value of adding new measurements to traditional disease parameters which could improve clinical decision-making about individual patients.

Stronger evidence from other cancers also shows how measuring miR-145-5p expression in various sample types holds promise as a diagnostic and/or prognostic biomarker for breast [91], lung [92], endometrial [93], glioblastoma [94], gastric [95], cervical [96] and urothelial [97] carcinomas. Together with our data, this evidence base warrants further investigation of miR-145-5p as a standalone or adjuvant biomarker for cancer, including prostate cancer. Critically, standardized approaches are fundamental for robust clinical evaluation of miRNA biomarkers [13].

Nonetheless, miR-145-5p alone likely lacks sufficient specificity and sensitivity to serve as a clinically useful biomarker. Instead, it would likely be included in a multivariate biomarker panel with other carefully selected genomic and proteomic markers. We have previously demonstrated the superiority of multivariate biomarker panels over individual markers for prostate cancer diagnosis and prognosis [98,99]. The synergistic combination of multiple miRNAs into a biomarker panel provides greater sensitivity and specificity than any individual miRNA biomarker alone (Appendix A). Given the miR-145-5p network highlighted here (Figure 5), a panel emphasizing EMT prediction could incorporate additional related genes or proteins, especially MYO6. In fact, current prostate cancer risk prediction models like STHLM3 incorporate various genomic, proteomic and clinical variables [100]. Our findings suggest miR-145-5p may be a valuable addition to such models. Considering its importance across cancers, miR-145-5p could have broad utility as a cancer biomarker [101].

Ultimately, clinical acceptance of miRNA biomarkers requires evidence that they improve patient management. The work described here is in part due to the need to have further understanding of miRNA biological functions in order to identify the most useful candidates, individually and in combination. Further obstacles to clinical adoption of miRNA biomarkers include concerns around reproducibility, lack of standardization across studies, and insufficient validation in large patient cohorts. Overcoming these barriers necessitates robust research, stringent validation protocols, and close collaboration between researchers, clinicians, and regulatory bodies. It is also important to acknowledge prostate tumor heterogeneity, as profiling is not cell-specific and the miR-145-5p/MYO6 relationship may be more significant in certain cell populations. Therefore, further work in this area is likely to incorporate single-cell analysis and advanced proteomics could provide insights into the cell-specific role of miR-145-5p to inform more precise diagnostics and targeted therapies [102,103].

## 4. Materials and Methods

### 4.1. Cell Culture and Transfections

All cell lines were acquired from the American Type Culture Collection (ATCC, Rockville, MD, USA). An in-house genotyping service authenticated the cells, and they were verified to be mycoplasma-free (InvivoGen, Toulouse, France). Cells utilized at a low passage number ranging from 3 to 6. RWPE-1, a normal prostate epithelial cell line, was cultured in keratinocyte growth medium, supplemented with 5 ng/mL of human recombinant epidermal growth factor and 0.05 mg/mL of bovine pituitary extract. (Life Technologies, Paisley, UK). The human prostate cancer cell lines, DU145 and PC3, were cultivated in RPMI-1640 medium, enriched with 10% fetal bovine serum and L-glutamine (Life Technologies). The cells were maintained at 37 °C under a humidified atmosphere of 95% air and 5% CO_2_. For miRNA transfections, in a 6-well plate, 1 × 10^5^ cells were seeded per well to ensure ~80% confluency at collection. Cells were transfected with 25 nM miR-145-5p precursor (hsa-miR-145-5p miRCURY LNA miRNA Mimic; ID YM0047001), miR-145-5p inhibitor (anti-miR-145-5p; ID YI04102423), or a 25 nM non-targeting negative control (Negative Control A; ID YI00199006) (Qiagen, Manchester, UK) using Lipofectamine 2000 (Life Technologies) for 4 to 6 h then replaced with culture media. After 72 h, cells were collected for RNA or protein extraction. Pre-miR-145-5p (double-stranded RNA molecule designed to mimic endogenous mature miR-145-5p) and anti-miR-145-5p (single stranded oligonucleotide designed to specifically bind and inhibit endogenous miR-145-5p) are chemically modified molecules designed upon the mature miR-145-5p sequence GUCCAGUUUUCCCAGGAAUCCCU.

### 4.2. Quantitative Real-Time PCR (qRT-PCR)

Total RNA was isolated from cell lines using the miRNeasy Tissue/Cells Advanced Mini Kit (Qiagen, Manchester, UK) following the manufacturer’s protocol. RNA integrity was verified by a NanoDropTM 2000 spectrophotometer (ThermoFisher Scientific, Waltham, MA, USA). First strand cDNA was synthesized from 500 ng total RNA using the Transcriptor First Strand cDNA Synthesis Kit (Roche, Sussex, UK) and random primers according to the manufacturer’s instructions. Quantitative PCR was performed using FastStart SYBR Green Master (Roche) on a Roche LC480 LightCycler with primer sets for:MYO6 (fw: GGATCTGTCCGAGCAGGAAG, rv: CTGTACGGGTGAAGCTGGAG),ACTA2 (fw: GTTCCGCTCCTCTCTCCAAC, rv: GTGCGGACAGGAATTGAAGC),VIM (fw: GGACCAGCTAACCAACGACA, rv: AAGGTCAAGACGTGCCAGAG),FN1 (fw: TCAGCTTCCTGGCACTTCTG, rv: TCCCTGGGGATGTGACCAAT), and housekeeping gene GAPDH (fw: GACAGTCAGCCGCATCTTCT, rv: GCGCCCAATACGACCAAATC).

Gene expression was normalized to GAPDH. The data presented were generated from at least three independent biological replicates. Quantitative reverse transcription PCR (qRT-PCR) for miR-145-5p was performed using the miRCURY LNA miRNA PCR Assays system (Qiagen). A total of 20 ng of template RNA was used in each first strand cDNA synthesis reaction. PCR amplification was carried out for 40 cycles and fluorescence was monitored using the Roche LC480 LightCycler. Normalization was performed against the snRNA U6 housekeeping gene. All qRT-PCR miRNA data were generated from a minimum of three independent biological replicates.

### 4.3. Protein Analysis

Protein extraction was performed using Cell Lysis Buffer (Abcam, Cambridge, UK) supplemented with 2% *v*/*v* HaltTM Protease Inhibitor Cocktail (ThermoFisher Scientific). Western blotting was conducted using Bio-Rad mini-Protean TGX gels and Trans-Blot^®^ Turbo Transfer System with associated reagents (Bio-Rad, Watford, UK). Primary antibodies used were rabbit anti-MYO6 and mouse anti-GAPDH as a loading control (both from Proteintech, Manchester, UK). Membranes were blocked with 5% milk in TBS-T (0.05%), followed by incubation with the appropriate horseradish peroxidase (HRP)-conjugated secondary antibody (goat anti-rabbit IgG-HRP at 1:5000 dilution or goat anti-mouse IgG-HRP at 1:5000 dilution, both from Proteintech). Chemiluminescent signal was detected using enhanced chemiluminescent reagent (ThermoFisher Scientific) and imaged on a G:BOX F3 system (Syngene, Cambridge, UK). A minimum of three biological replicates were performed for each experiment.

### 4.4. Bioassays

For proliferation assays, transfected cells were replated at 0.01 × 10^6^ cells per well. After 24 h, 10% *v*/*v* alamarBlue^TM^ Cell Viability Reagent was added and incubated at 37 °C in a humidified atmosphere of 95% air and 5% CO_2_ for 4 h. Following incubation, absorbance was measured at 570 nm and 600 nm using a FLUOstar Omega plate reader (BMG LabTech, Aylesbury, UK). For migration assays, transfected cells were replated at 0.25 × 10^6^ cells per well and allowed to form a confluent monolayer. A 200 μL pipette tip was used to create an artificial “wound”. Wound images were captured at 0 and 72 h. Percentage closure was calculated as ((initial gap area − remaining gap area)/initial gap area) × 100. For colony formation assays, transfected cells were replated at 0.03 × 10^6^ cells per well and incubated for 72 h at 37 °C in a humidified atmosphere of 95% air and 5% CO_2_. Cells were quantified using the cell counting function in ImageJ Version 1.54h software. Fold change was calculated relative to the non-transfected control (NTC).

### 4.5. Databases and Analysis

The prostate adenocarcinoma (PRAD) data from The Cancer Genome Atlas (TCGA) repository was obtained from the Genomic Data Commons Data Portal v39.0 (https://portal.gdc.cancer.gov/, accessed on 10 January 2024). Expression analysis of miR-145-5p and correlation with clinical parameters were performed using the University of California Santa Cruz (UCSC) Xena Functional Genomics Explorer (https://xenabrowser.net/, accessed on 12 January 2024) [104]. Identification of negatively correlated targets and functional enrichment analysis were conducted using CancerMIRNome incorporating clusterProfiler 1.0 (http://bioinfo.jialab-ucr.org/CancerMIRNome/, accessed on 28 July 2023) [105,106]. Epithelial-mesenchymal transition (EMT)-associated targets of miR-145-5p were determined by integrating data from miRTarBase v9.0 (http://mirtarbase.cuhk.edu.cn/, accessed on 14 February 2023) [107], the EMT gene database dbEMT 2.0 (http://dbemt.bioinfo-minzhao.org/, accessed on 16 February 2023) [108], EMTome (http://www.emtome.org/, accessed on 28 July 2023) [109], and Venny 2.0 (https://bioinfogp.cnb.csic.es/tools/venny/, accessed on 28 July 2023) [110] tools. The protein–protein interaction network of MYO6 was generated using STRING v12.0 (https://string-db.org/, accessed on 24 March 2023) [111] and functionally annotated using Kyoto Encyclopedia of Genes and Genomes (KEGG) pathways. Additional survival analyses were performed with the Kaplan–Meier Plotter (KM-Plotter) (http://kmplot.com/analysis/, accessed on 25 June 2023) [112]. Network analysis and visualization were conducted using GeneMANIA (https://genemania.org/, accessed on 21 April 2023) [113] and miRTargetLink 2.0 (http://ccbcompute.cs.uni-saarland.de/mirtargetlink2, accessed on 22 April 2023) [114]. Combination biomarker panels accessed using CombiROC (http://CombiROC.eu, accessed on 21 April 2023) [115]. Hallmark enrichment analysis performed with CancerHallmark Tool (https://cancerhallmarks.com/, accessed on 10 March 2024).

### 4.6. Statistics

Graphs were generated using GraphPad PRISM version 9. Unless otherwise stated, bar graphs show mean ± standard error of at least three biological replicates, with statistical significance evaluated by paired *t*-test. All boxplots show mean and Tukey whiskers, with statistical significance determined by unpaired *t*-test with Welch’s correction or nonparametric Kruskal–Wallis one-way ANOVA with Dunn’s multiple comparisons test. Statistical significance for scatterplots was evaluated by Pearson’s correlation with *p*-values adjusted for multiple hypothesis testing. For Kaplan–Meier graphs, statistical significance was determined by log-rank (Mantel-Cox) test. To correct for multiple hypotheses in functional enrichment tables, adjusted *p*-values were calculated using the Benjamini-Hochberg procedure. For all analyses, data were considered statistically significant at * *p* < 0.05, ** *p* < 0.01, *** *p* < 0.001, and **** *p* < 0.0001.

## 5. Conclusions

Our findings demonstrate that miR-145-5p expression is decreased in prostate cancer and correlates with indicators of disease advancement. To our knowledge, this is the first evidence that miR-145-5p directly regulates MYO6 in prostate cancer cells. We hypothesize that declining miR-145-5p levels may promote EMT in prostate cancer. Further research should explore the miR-145-5p/MYO6 interaction and its role in EMT. MiR-145-5p has potential as a diagnostic or prognostic biomarker for prostate cancer, warranting additional investigation.

## Figures and Tables

**Figure 1 ijms-25-04301-f001:**
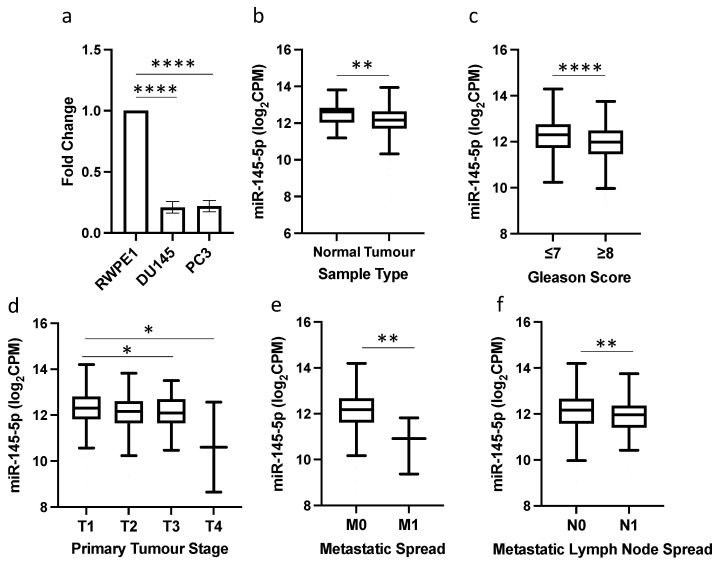
miR-145-5p downregulation is associated with prostate cancer and disease progression. (**a**) qRT-PCR shows miR-145-5p expression is significantly lower in DU145 and PC3 prostate cancer cell lines compared to normal prostate cell line, RWPE1 (*n* = 3, housekeeping: snRNA U6). (**b**) UCSC Xena analysis of TCGA PRAD samples shows miR-145-5p expression is significantly decreased in prostate tumor tissue (*n* = 494) compared to normal prostate tissue (*n* = 52). UCSC Xena analysis of TCGA PRAD samples shows expression of miR-145-5p is significantly lower in patients with (**c**) Gleason score ≥ 8 (*n* = 210) compared to those scored ≤ 7 (*n* = 336), (**d**) pathological stage T3 (*n* = 55) and T4 (*n* = 2) compared to T1 (*n* = 197) (**e**) pathological stage M1 (*n* = 3) compared to M0 (*n* = 542) and (**f**) pathological stage N1 (*n* = 80) compared to N0 (*n* = 385). All *p*-values generated by unpaired two-tailed *t*-test (* *p* < 0.05, ** *p* < 0.01, **** *p* < 0.0001). CPM = Copies per million; *n* = number.

**Figure 2 ijms-25-04301-f002:**
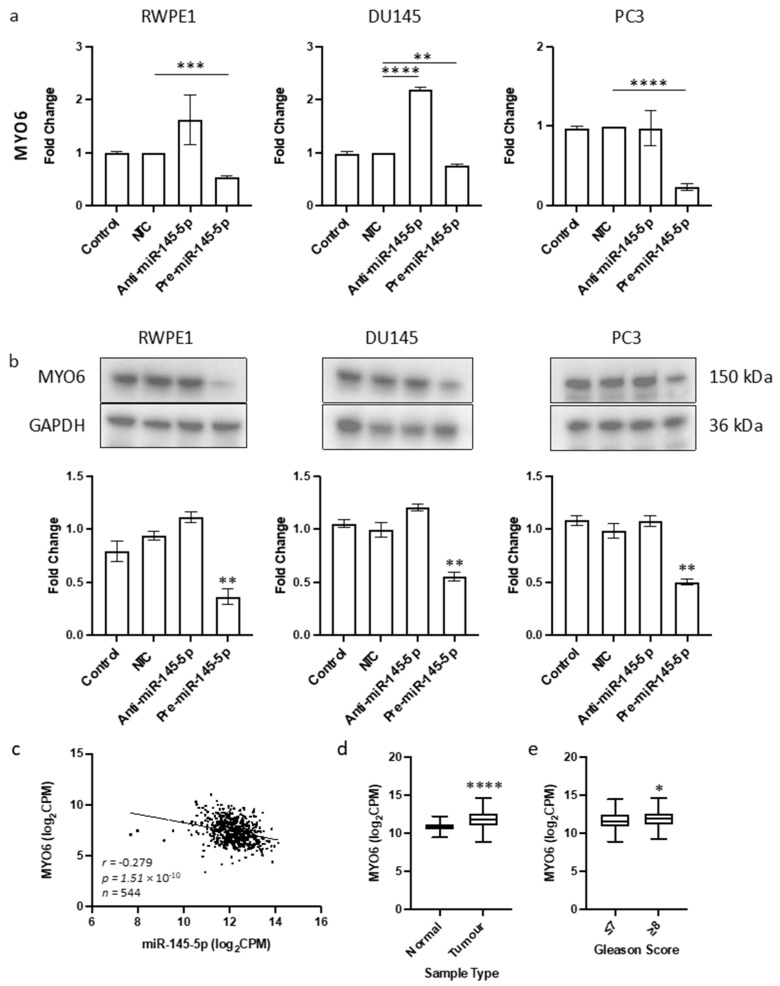
Identification of MYO6 as novel target of miR-145-5p in prostate cancer. (**a**) Transient overexpression of miR-145-5p significantly reduces MYO6 gene expression in RWPE1, DU145 and PC3 cell lines. (*n* = 3, expression values normalized to GAPDH) (**b**) Quantified Western blotting and representative images (*n* = 3) shows transient overexpression of miR-145-5p causes significant downregulation of MYO6 protein in RWPE1, DU145 and PC3 cell lines. Bar graphs show mean ± SEM. *p*-values generated by unpaired two-tailed *t*-test, relative to NTC (* *p* < 0.05, ** *p* < 0.01, *** *p* < 0.001, **** *p* < 0.0001). (**c**) CancerMIRNome analysis of the TCGA PRAD specimens (*n* = 543) shows the expression of miR-145-5p and MYO6 are significantly negatively correlated (Pearson correlation, **** *p* < 0.0001). UCSC Xena analysis of TCGA PRAD samples shows MYO6 expression is significantly elevated in (**d**) tumor tissue (*n* = 497) relative to normal (*n* = 52) tissue and in (**e**) samples with Gleason score ≥ 8 (*n* = 154) compared to those scored ≤ 7 (*n* = 213). (both Welch’s *t*-test, (* *p* < 0.05, **** *p* < 0.0001). NTC = Non-Targeting Control; *n* = number.

**Figure 3 ijms-25-04301-f003:**
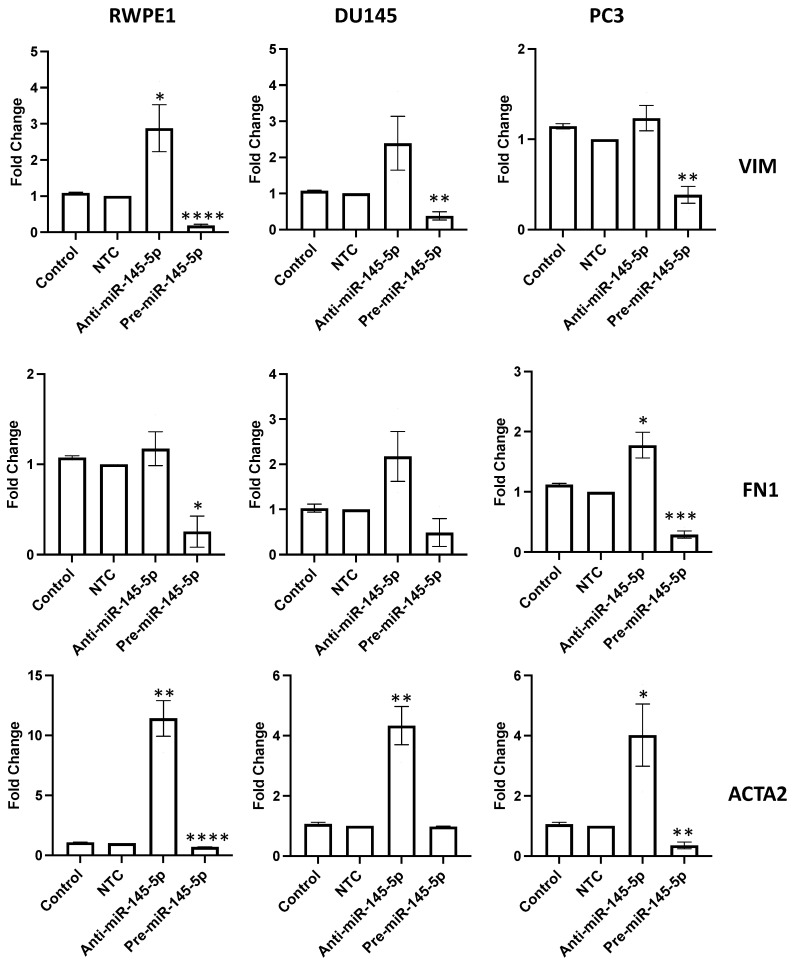
miR-145-5p alters the expression of key EMT markers. VIM, FN1 and ACTA2 quantified 72 h post transfection using qRT-PCR. Bars show mean ± SEM (*n* = 3, expression values normalized to GAPDH). *p*-values generated by unpaired two-tailed *t*-test, relative to NTC (* *p* < 0.05, ** *p* < 0.01, *** *p* < 0.001, **** *p* < 0.0001). NTC = Non-Targeting Control; *n* = number.

**Figure 4 ijms-25-04301-f004:**
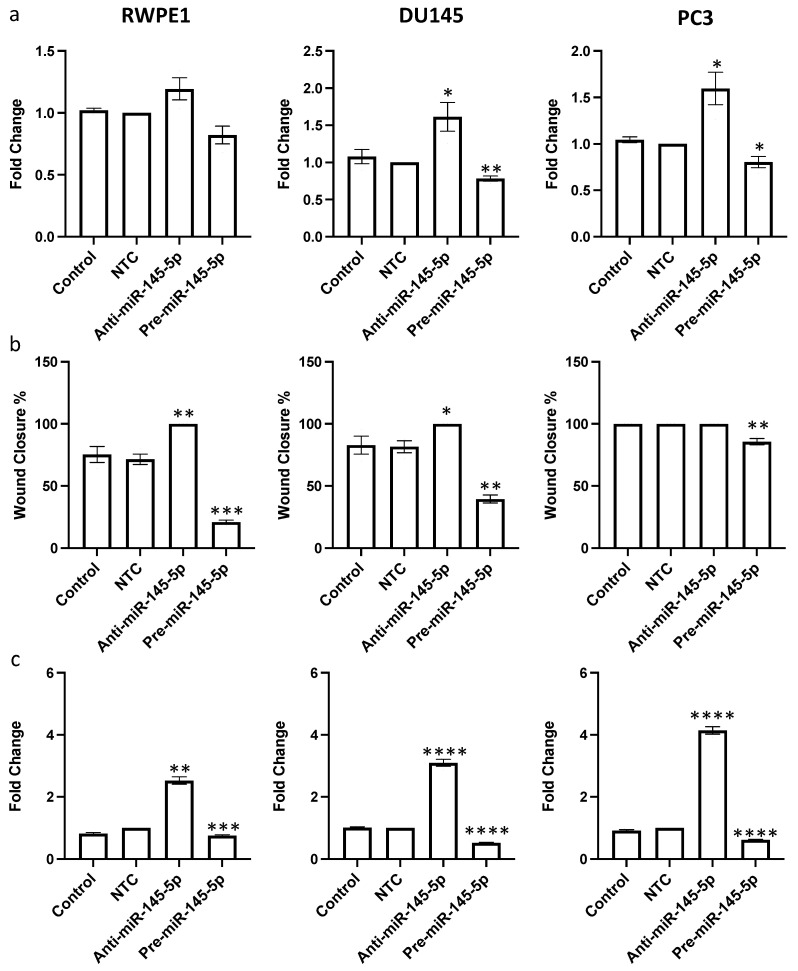
miR-145-5p alters (**a**) proliferation, (**b**) migration and (**c**) clonogenicity of RWPE1, DU145 and PC3 cells. Proliferation measured by Alamar blue absorbance (600 nm) 72 h post transfection (*n* = 5). Migration measured by scratch assay performed at 72 h post transfection using ImageJ to quantify wound closure (*n* = 3). Clonogenicity measured by colony counting performed at 72 h post transfection using ImageJ (Version 1.54h) to quantify. Bar graphs show mean ± SEM. *p*-values generated by unpaired two-tailed *t*-test, relative to NTC (* *p* < 0.05, ** *p* < 0.01, *** *p* < 0.001, **** *p* < 0.0001). NTC = Non-Targeting Control; *n* = number.

**Figure 5 ijms-25-04301-f005:**
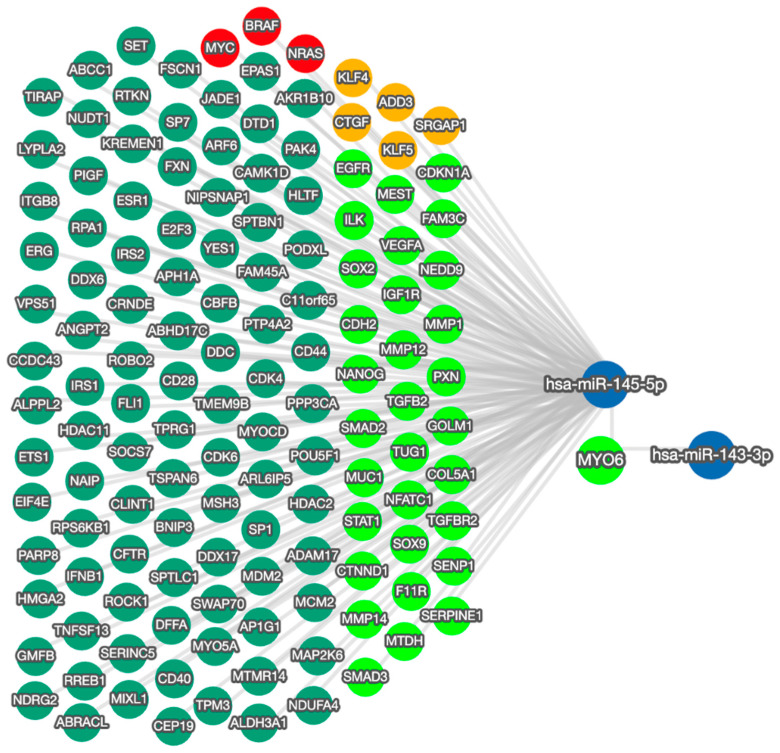
miRTargetLink 2.0 visualization of miR-145-5p and MYO6 bidirectional network interactions. miR-145-5p and MYO6 were input items, while the connected nodes are gene/miRNA interactions validated by strong experimental evidence (qRT-PCR, Western blot, Luciferase reporter assay). Blue nodes: miRNAs; green nodes: genes; bright green nodes: EMT-associated genes; red nodes: genes significantly associated with prostate cancer; orange nodes: actin cytoskeleton modulators.

**Figure 6 ijms-25-04301-f006:**
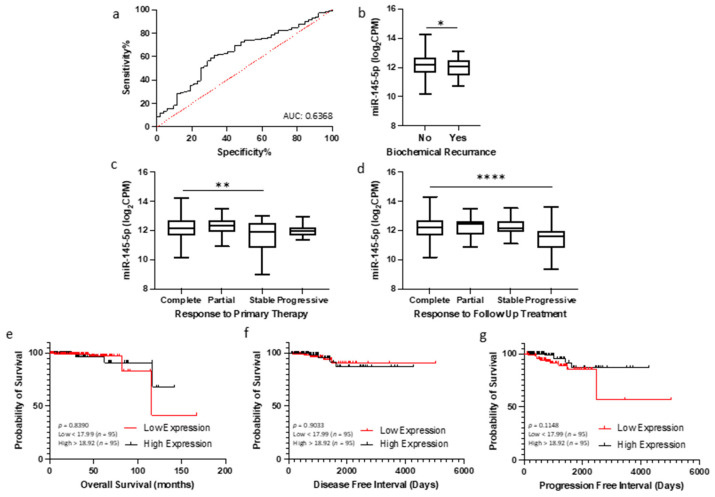
miR-145-5p as a prognostic biomarker for prostate cancer. (**a**) ROC curve analysis demonstrating that miR-145-5p shows potential for distinguishing between tumor and normal tissue in TCGA PRAD patient cohort. (**b**) Patients experiencing biochemical recurrence (BCR) showed significantly lower levels of miR-145-5p compared to those with no recurrence (*p*-values generated by unpaired two-tailed *t*-test; *n*, no recurrence = 407, recurrence = 61). (**c**) Significant difference in miR-145-5p levels between patient remission response after primary therapy (*n*, complete = 330, partial = 30, stable = 29, progressive = 23) and (**d**) follow up therapy (*n*, complete = 340, partial = 17, stable = 42, progressive = 24)(* *p* < 0.05, ** *p* < 0.01, **** *p* < 0.0001). For KM plots, patients were divided into quartiles based on miR-145-5p expression for (**e**) Overall survival, (**f**) Disease-free interval and (**g**) Progression-free interval. (*p*-values generated by log-rank (Mantel–Cox) test). ROC = Receiver operating characteristic. AUC = Area Under Curve. *n* = number.

**Table 1 ijms-25-04301-t001:** Functional enrichment analysis of miR-145-5p in prostate cancer. Table shows the significant association of miR-145-5p target genes with Gene Set descriptions related to prostate cancer. Analysis performed using cluster Profiler in CancerMIRNome.

GENE SET	GENE SET ID	Description	Count/ List Total	Adjusted *p*-Value ^1^	Gene Symbol
KEGG	hsa05215	Prostate cancer	9/238	3.37 × 10^−4^	*CDKN1A*; *EGFR*; *IGF1R*; *ERG*; *NRAS*; *MDM2*; *BRAF*; *PDGFD*; *E2F3*
Disease Ontology	DOID:10283	prostate cancer	29/238	7.82 × 10^−8^	*MUC1*; *MYO6*; *CDKN1A*; *IRS1*; *EGFR*; *MYC*; *IFNB1*; *IGF1R*; *VEGFA*; *SERPINE1*; *ESR1*; *NUDT1*; *MDM2*; *MMP1*; *MMP12*; *MMP14*; *SMAD3*; *SMAD4*; *PDGFD*; *CTNND1*; *SP1*; *RPS6KA3*; *IGFBP5*; *MUC4*; *ABCC1*; *SET*; *HIF1A*; *PXN*; *MSH3*
	DOID:10286	prostate carcinoma	7/238	3.15 × 10^−2^	*EGFR*; *MYC*; *IGF1R*; *VEGFA*; *ESR1*; *MMP14*; *PDGFD*
DisGeNET	umls:C0936223	Metastatic Prostate Carcinoma	13/238	6.37 × 10^−5^	*KLF4*; *MUC1*; *CDKN1A*; *EGFR*; *MYC*; *VEGFA*; *JAG1*; *ERG*; *MMP14*; *CD44*; *CTNND1*; *SENP1*; *HIF1A*
	umls:C0007112	Adenocarcinoma of prostate	11/238	6.54 × 10^−5^	*EGFR*; *VEGFA*; *SERPINE1*; *ESR1*; *ERG*; *ILK*; *CD44*; *BRAF*; *PSAT1*; *DUSP6*; *HIF1A*
	umls:C1654637	androgen independent prostate cancer	10/238	2.58 × 10^−4^	*CDKN1A*; *PPP3CA*; *EGFR*; *FSCN1*; *ESR1*; *ERG*; *ADAM17*; *PTP4A2*; *PSAT1*; *HIF1A*
	umls:C0278838	Prostate cancer recurrent	3/238	2.77 × 10^−2^	*EGFR*; *SOX9*; *PSAT1*
	umls:C1328504	Hormone refractory prostate cancer	4/238	4.79 × 10^−2^	*EGFR*; *FSCN1*; *ESR1*; *PSAT1*

KEGG = Kyoto Encyclopaedia of Genes and Genomes; ^1^ Adjusted *p*-value for multiple hypothesis correction used Benjamini and Hochberg procedure.

**Table 2 ijms-25-04301-t002:** Functional enrichment analysis of EMT-related functions associated with miR-145-5p. Table shows the significant association of miR-145-5p target genes with Gene Set descriptions related to EMT. Analysis performed using cluster Profiler in CancerMIRNome.

GENE SET	GENE SET ID	Description	Count/ List Total	Adjusted *p*-Value ^1^	Gene Symbol
KEGG	hsa04350	TGF-beta signaling pathway	11/238	7.52 × 10^−6^	*MYC*; *SMAD3*; *SMAD5*; *TGFBR2*; *SMAD4*; *SP1*; *ZFYVE9*; *ROCK1*; *RPS6KB1*; *TGFB2*; *SMAD2*
	hsa04520	Adherens junction	8/238	2.37 × 10^−4^	*YES1*; *EGFR*; *IGF1R*; *ACTB*; *SMAD3*; *TGFBR2*; *SMAD4*; *CTNND1*
	hsa04510	Focal adhesion	12/238	1.18 × 10^−3^	*EGFR*; *IGF1R*; *VEGFA*; *ITGB8*; *PAK4*; *ILK*; *BRAF*; *ACTB*; *PDGFD*; *ROCK1*; *TNR*; *PXN*
REACTOME	R-HSA-1474244	Extracellular matrix organization	12/238	3.14 × 10^−2^	*SERPINE1*; *ITGB8*; *ADAM17*; *F11R*; *MMP1*; *MMP12*; *MMP14*; *COL5A1*; *CD44*; *P4HA1*; *TNR*; *TGFB2*
	R-HSA-446728	Cell junction organization	6/238	3.28 × 10^−2^	*ILK*; *CDH2*; *F11R*; *ACTB*; *CTNND1*; *PXN*
	R-HSA-1442490	Collagen degradation	5/238	3.44 × 10^−2^	*ADAM17*; *MMP1*; *MMP12*; *MMP14*; *COL5A1*
GO-BP	GO:0010810	regulation of cell-substrate adhesion	15/238	1.38 × 10^−5^	*FZD7*; *VEGFA*; *SERPINE1*; *JAG1*; *NEDD9*; *ILK*; *MMP12*; *MMP14*; *RREB1*; *SMAD3*; *CDK6*; *ANGPT2*; *ROCK1*; *CCDC80*; *WASHC2C*
	GO:0031589	cell-substrate adhesion	19/238	1.45 × 10^−5^	*FZD7*; *VEGFA*; *SERPINE1*; *JAG1*; *NEDD9*; *ILK*; *CTGF*; *MMP12*; *MMP14*; *RREB1*; *CD44*; *SMAD3*; *CDK6*; *ANGPT2*; *ROCK1*; *CCDC80*; *MUC4*; *PXN*; *WASHC2C*
	GO:0010464	regulation of mesenchymal cell proliferation	7/238	1.99 × 10^−5^	*STAT1*; *MYC*; *VEGFA*; *IRS2*; *SOX9*; *TGFBR2*; *CTNNBIP1*
	GO:0048762	mesenchymal cell differentiation	14/238	4.83 × 10^−5^	*STAT1*; *JAG1*; *DDX17*; *HDAC2*; *SOX9*; *SOX11*; *SMAD3*; *TGFBR2*; *SMAD4*; *ERBB4*; *HMGA2*; *HIF1A*; *TGFB2*; *SMAD2*
	GO:0010632	regulation of epithelial cell migration	16/238	5.21 × 10^−5^	*KLF4*; *MAP3K3*; *VEGFA*; *ADAM17*; *ETS1*; *RREB1*; *SOX9*; *TGFBR2*; *SP1*; *ARF6*; *ANGPT2*; *SRPX2*; *HBEGF*; *HIF1A*; *CD40*; *TGFB2*
	GO:0001837	epithelial to mesenchymal transition	11/238	8.39 × 10^−5^	*JAG1*; *DDX17*; *HDAC2*; *SOX9*; *SMAD3*; *TGFBR2*; *SMAD4*; *HMGA2*; *HIF1A*; *TGFB2*; *SMAD2*
	GO:0010631	epithelial cell migration	17/238	9.64 × 10^−5^	*KLF4*; *MAP3K3*; *VEGFA*; *ADAM17*; *ETS1*; *RREB1*; *SOX9*; *TGFBR2*; *SP1*; *ARF6*; *ANGPT2*; *SRPX2*; *HBEGF*; *HIF1A*; *PXN*; *CD40*; *TGFB2*
	GO:0001952	regulation of cell-matrix adhesion	9/238	4.96 × 10^−4^	*VEGFA*; *SERPINE1*; *JAG1*; *ILK*; *MMP12*; *MMP14*; *SMAD3*; *CDK6*; *ROCK1*
MSigDB	HALLMARK_EMT	HALLMARK_EMT	11/238	4.15 × 10^−2^	*VEGFA*; *SERPINE1*; *CTGF*; *CDH2*; *MEST*; *MMP1*; *MMP14*; *COL5A1*; *CD44*; *SNTB1*; *TGFBI*

KEGG = Kyoto Encyclopaedia of Genes and Genomes; GO-BP = Gene Ontology–Biological Process; MSigDB = Molecular Signatures Database; ^1^ Adjusted *p*-value for multiple hypothesis correction used Benjamini and Hochberg procedure.

## Data Availability

The genotypic and phenotypic data for Prostate adenocarcinoma (PRAD) cohort are available at The Cancer Genome Atlas (TCGA) portal. Analysis tools are listed in Methods and other datasets analyzed in the present study are available from the published papers that have been cited in this manuscript.

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
