# Peer review of "The Suppression of the Epithelial to Mesenchymal Transition in Prostate Cancer through the Targeting of MYO6 Using MiR-145-5p"

_ijms, 2024, doi:10.3390/ijms25084301_

Round 1

Reviewer 1 Report

Comments and Suggestions for Authors

This study explores the novel aspect of miR-145-5p in prostate cancer, particularly its role in regulating EMT and interaction with MYO6 gene, areas of significant interest in the field.

The combined use of in silico and in vitro analyses provides a solid foundation for understanding the molecular mechanisms and potential clinical implications of miR-145-5p. The incorporation of The Cancer Genome Atlas (TCGA PRAD) data notably strengthens the study's findings with a substantial dataset.

The manuscript showcases a well-executed study that captivates readers with its clarity and relevance. It provides a detailed exploration of miR-145-5p's role in the development and progression of prostate cancer, highlighting its therapeutic and biomarker potential. The authors offer extensive methodologies, comprehensive results supported by robust data, and well-crafted figures, followed by a thorough discussion that situates the findings within a broader scientific framework.

Reviewing this article was a pleasure, marked by an appreciation for the clear presentation of findings and the logical progression of arguments and discussions.

Nevertheless, it's still worth considering whether the insights gained from cell line studies and bioinformatic analyses are directly applicable to clinical settings and patient outcomes. Moreover, addressing the practical hurdles in the clinical application of these findings, such as the implications of employing miR-145-5p as a biomarker in prostate cancer, and the strategies for targeting or modulating miR-145-5p in prostate cancer treatments, represents a vital direction for future research. 

Taken together, I am pleased with the quality of this manuscript and look forward to the opportunity to review additional research in this promising field.

Author Response

REVIEWER 1

We are delighted that this reviewer found the paper a pleasure to read. We highlight one comment from the review below which we felt pertinent to address.

“Nevertheless, it's still worth considering whether the insights gained from cell line studies and bioinformatic analyses are directly applicable to clinical settings and patient outcomes. Moreover, addressing the practical hurdles in the clinical application of these findings, such as the implications of employing miR-145-5p as a biomarker in prostate cancer, and the strategies for targeting or modulating miR-145-5p in prostate cancer treatments, represents a vital direction for future research.”

We entirely agree with the reviewer that implementation of a new biomarker like miR-145-5p is likely to face hurdles to clinical adoption. We have touched upon this briefly in discussion, but in light of comment above, we now add further context for this to fully emphasize the need for well-designed, robust studies to help overcome these hurdles and realise the clinical potential of miRNA biomarkers (line 412-416) 

SUMMARY OF CHANGES

Thank you for the review of the above paper. We appreciate the time taken by the reviewers to read our manuscript and we are pleased they recognize the potential importance and value of our work. We thank the reviewers for their insightful comments, which have helped to improve the paper.

In response to the comments made by the reviewers, we have amended our manuscript accordingly and tracked the changes. A TRACKED and CLEAN version of revised manuscript are provided.

Each reviewer comment has been numbered separately below and are addressed in turn in our responses.

Where our responses contain references to either specific figures/tables in the manuscript or specific sections of the text, the line number position of these sections have been clearly indicated. These lines refer to the numbering in the CLEAN manuscript and are highlighted in red text

We have also corrected some minor typographical errors in the process of reviewing our manuscript.

We trust you find these revisions satisfactory.

Reviewer 2 Report

Comments and Suggestions for Authors

The author study the role of miR145-5p in EMT using a prostate cancer cell model. The experiments are well thought of and the controls are suitable. The paper is well written and comprehensible for most readers.

The only suggestion I make is that many authors hav e shown a direct correlation between EMT and chemoresistance in cancer. Many authors have show, in fact (even using the PC-3 cell line), that inducing resistance can lead to EMT changes and that inhibiting EMT can lead to increaed sensitivity to anticancer drugs. The papaer would greatly benefit from a sigle simple experiment demonstrating if the changes in expression of the microRNA can lead to changes in drug sensitivity.

Author Response

REVIEWER 2

The only suggestion I make is that many authors have shown a direct correlation between EMT and chemoresistance in cancer. Many authors have show, in fact (even using the PC-3 cell line), that inducing resistance can lead to EMT changes and that inhibiting EMT can lead to increaed sensitivity to anticancer drugs. The papaer would greatly benefit from a sigle simple experiment demonstrating if the changes in expression of the microRNA can lead to changes in drug sensitivity.

We thank the reviewer for this interesting suggestion. As they note, other studies have already examined the link between miR-145 expression and chemosensitivity in prostate cancer cells, so it was beyond the scope of this paper to simply replicate those experiments. Instead, the primary focus of this paper was to evaluate the biomarker potential of miR-145-5p in prostate cancer, rather than its value as a therapeutic target, as we feel miRNA profiling is a more likely to be translated into clinical use in the near future. However, we agree that investigation of changing cell chemosensitivity is a very valuable suggestion for future research that wants to demonstrate the potential therapeutic potential of modulating miR-145-5p expression. We have therefore included new sentences in the discussion to remind readers that this a route for further investigation (lines 326-330)

SUMMARY OF CHANGES

Thank you for the review of the above paper. We appreciate the time taken by the reviewers to read our manuscript and we are pleased they recognize the potential importance and value of our work. We thank the reviewers for their insightful comments, which have helped to improve the paper.

In response to the comments made by the reviewers, we have amended our manuscript accordingly and tracked the changes. A TRACKED and CLEAN version of revised manuscript are provided.

Each reviewer comment has been numbered separately below and are addressed in turn in our responses.

Where our responses contain references to either specific figures/tables in the manuscript or specific sections of the text, the line number position of these sections have been clearly indicated. These lines refer to the numbering in the CLEAN manuscript and are highlighted in red text

We have also corrected some minor typographical errors in the process of reviewing our manuscript.

We trust you find these revisions satisfactory.